# Comprehensive Organ-Specific Profiling of Douglas Fir (*Pseudotsuga menziesii*) Proteome

**DOI:** 10.3390/biom13091400

**Published:** 2023-09-16

**Authors:** Caroline Teyssier, Odile Rogier, Stéphane Claverol, Florian Gautier, Marie-Anne Lelu-Walter, Harold Duruflé

**Affiliations:** 1INRAE, ONF, BioForA, UMR 0588, 45075 Orleans, France; 2Plateforme de Protéomique, Université de Bordeaux, 33405 Bordeaux, France

**Keywords:** *Pseudotsuga menziesii*, proteomics, shotgun, multi-organ

## Abstract

The Douglas fir (*Pseudotsuga menziesii*) is a conifer native to North America that has become increasingly popular in plantations in France due to its many advantages as timber: rapid growth, quality wood, and good adaptation to climate change. Tree genetic improvement programs require knowledge of a species’ genetic structure and history and the development of genetic markers. The very slow progress in this field, for Douglas fir as well as the entire genus *Pinus*, can be explained using the very large size of their genomes, as well as by the presence of numerous highly repeated sequences. Proteomics, therefore, provides a powerful way to access genomic information of otherwise challenging species. Here, we present the first Douglas fir proteomes acquired using nLC-MS/MS from 12 different plant organs or tissues. We identified 3975 different proteins and quantified 3462 of them, then examined the distribution of specific proteins across plant organs/tissues and their implications in various molecular processes. As the first large proteomic study of a resinous tree species with organ-specific profiling, this short note provides an important foundation for future genomic annotations of conifers and other trees.

## 1. Introduction

Due to its rapid growth and favorable wood quality, Douglas fir (*Pseudotsuga menziesii* Mirb.) is an economically important species for timber production, supplying about 70% of the demand for softwoods in the United States. On good plantation sites, maximum tree heights can reach 30 m at 40 years, with up to 800 m^3^ in volume. Douglas fir production is also increasing in Europe [1], as this species is resistant to many European pathogens, and it is expected that Douglas fir will be better adapted to future climate conditions in Central Europe than any other species [2]. In order to ensure the success of future plantations, knowledge of Douglas fir genetics will be important for selecting plantation stock.

Despite the economic and ecological importance of several conifer species in the family *Pinaceae*, progress in the area of whole genome sequencing and associated marker development has been limited [3]. The genome sizes of conifers are larger than those of most other plant species, and the genus *Pinus* has particularly large and complex genomes [4], with estimated sizes that range between 19 Gbp for the Norway spruce (*Picea abies*) and 35 Gbp for the sugar pine (*Pinus lambertiana*) [5,6]. Such large genomes, combined with high heterozygosity and numerous repetitive DNA elements, present major obstacles to sequencing the genomes of many conifer species [7,8,9,10].

Because of these challenges, previous research in conifer genomics has focused largely on the transcribed portions of genomes. Transcriptome assemblies resulting from large-scale expressed sequence tag (EST) collections in conifers provide important genomic resources [11]. This technique facilitates the analysis of the transcribed part of the genome, which is not easy to predict from the entire genome. Only a few transcriptomes for several conifer species have been recently published, including *Pinus cembra* [12], *P. tabuliformis* [13], *P. halepensis* [14], *P. contorta*, and *P. banksiana* [15], and *Pseudotsuga menziesii* [16], and some exome capture studies have been performed with natural populations of *Pinus taeda* [17,18].

Some *Pinaceae* species genomes have been sequenced, like *Picea abies* [19], *Picea glauca* [20], and *Pinus taeda* [21]. The first reference genome sequence for Douglas fir was published recently [16] and highlights large anatomical, morphological, and physiological differences between angiosperms and gymnosperms while beginning to reveal the evolutionary changes that have occurred within the family *Pinaceae*.

In addition to transcriptomic studies, proteomics provides another promising approach to obtaining genomic information [22]. Mainly in the past, proteomic studies were carried out based on 2-DE gels associated with mass spectrometry identification of the proteins to whom the spot volume changed between the two compared conditions, leading to a limited number of identified proteins [23,24]. Shotgun proteomics—the combination of iTRAQ-based and LC–MS/MS methods, provides an effective tool for high-performance proteomic studies that can lead to whole proteome identification [25,26,27]. However, despite technological advancements in proteomics and the relatively lower cost of obtaining a complete proteome compared to transcriptome or genome sequencing, proteomics is still almost exclusively used to compare proteomes present under different conditions, looking only at proteins that are differentially abundant under compared conditions [28,29,30,31]. The experimental observation of proteins in a large set of organs makes it possible to validate the presence of proteins initially predicted using genome bioinformatics approaches. However, a few studies present shotgun proteomic data from different plant organs: seeds (in *Pinus greggii* and *Pinus patula* [32]), leaves (*Pinus radiata* and *Quercus ilex* [33]), shoot apical meristem (*Cunninghamia lanceolata* [34]), different stages of somatic embryos (*Larix principis-rupprechtii* [35]); as well as a cells fraction with this study of nuclear proteome of seedling (*Pinus radiata* [36]) and or of cell wall proteins of *Arabidopsis* mature stems [37]. Additional proteomic studies have been conducted for some Quercus species that also include analysis of many organs [38]. 

Here, we present results of shotgun proteomic analyses of 12 organs or tissues from Douglas fir: root, stem, xylem, needle, bud, female and male strobili, immature and mature seed, immature and mature somatic embryos, and callus. Organ-specific proteins and biological processes are presented and discussed. 

## 2. Materials and Methods

### 2.1. Plant Material and Sampling

Douglas fir buds, needles, and strobili were collected on 13 April 2018; immature seeds were collected on 23 July 2018; and mature seeds were collected on 30 August 2018. Mature somatic embryos were obtained by cultivation on Glitz maturation medium, using embryogenic line TD1s initiated from immature zygotic embryos originating from controlled cross [39]. The embryogenic mass with immature somatic embryos and non-embryogenic callus lines were issues of embryonal mass obtained using a reiteration of the somatic embryogenesis initiation [39]. Details of all the genotypes used and harvested are presented in Appendix A. Accurately weighed embryogenic and non-embryogenic samples were placed in 2 mL Eppendorf tubes after collection into Petri dishes. All other samples were kept after collection in a plastic bag on ice until cut to size and accurately weighed in 2 mL Eppendorf tubes. All tubes were immediately frozen in liquid nitrogen and stored at −80 °C until protein extraction. 

### 2.2. Sample Preparation and Protein Digestion 

Total protein extracts were prepared from three biological replicates of each sample (150 mg fresh mass) with a trichloroacetic acid (TCA)-acetone extraction. Proteins were re-suspended in a solubilization medium of 7M urea, 2M thiourea, 4% CHAPS (*v*/*v*), 10 mM dithiothreitol, 0.4% Triton X100 (*v*/*v*), and 80 mM Tris HCl pH 6.8 [40]. Protein samples were solubilized in Laemmli buffer, and 10 µg per sample was deposited onto SDS-PAGE gel. After colloidal blue staining, bands were cut in 1 mm^2^ gel pieces, destained in 25 mM ammonium bicarbonate 50% acetonitrile, rinsed twice in ultrapure water, and shrunk in acetonitrile. After acetonitrile removal, gel pieces were dried and incubated overnight at 37 °C in a trypsin solution (10 ng/µL in 50 mM NH_4_HCO_3_). Then, the pieces were incubated for 15 min in 50 mM NH_4_HCO_3_ at room temperature. The supernatant was collected, and an H_2_O/acetonitrile/HCOOH (47.5:47.5:5) extraction solution was added onto gel slices for 15 min. The extraction step was repeated twice and supernatants were pooled and dried in a vacuum centrifuge. Digests were then solubilized in 0.1% HCOOH.

### 2.3. nLC-MS/MS Analysis and Label-Free Quantitative Data Analysis

Peptide mixtures were analyzed on an Ultimate 3000 nanoLC system (Dionex, Amsterdam, The Netherlands) coupled to an Electrospray Orbitrap Fusion™ Lumos™ Tribrid™ Mass Spectrometer (Thermo Fisher Scientific, San Jose, CA, USA). Ten microliters of peptide digests were loaded onto a 300-µm-inner diameter x 5 mm C_18_ PepMap^TM^ trap column (LC Packings) at a flow rate of 10 µL/min. The peptides were eluted from the trap column onto an analytical 75 mm id × 50 cm C18 Pep-Map column (LC Packings) with a 4–40% linear gradient of solvent B in 48 min (solvent A was 0.1% formic acid and solvent B was 0.1% formic acid in 80% acetonitrile). The separation flow rate was set at 300 nL/min. The mass spectrometer operated in positive ion mode at a 1.9-kV needle voltage. Data were acquired using Xcalibur 4.3 software in a data-dependent mode. MS scans (*m/z* 375–1500) were recorded at a resolution of R = 120,000 (@ *m*/*z* 200) and an AGC target of 4 × 10^5^ ions collected within 50 ms. Dynamic exclusion was set to 60 s and top speed fragmentation in HCD (Higher-energy Collisional Dissociation) mode was performed over a 3 s cycle. MS/MS scans with a target value of 3 × 10^3^ ions were collected in the ion trap with a maximum fill time of 300 ms. Only +2 to +7 charged ions were selected for fragmentation. Other settings were as follows: heated capillary temperature, 275 °C; normalized HCD collision energy of 30%, and an isolation width of 1.6 *m*/*z*. Monoisotopic precursor selection (MIPS) was set to “Peptide” and an intensity threshold was set to 5 × 10^3^. The mass spectrometry proteomics data have been deposited to the ProteomeXchange Consortium via the PRIDE [41] partner repository with the dataset identifier PXD037815.

### 2.4. Database Search and Results Processing

Data were searched by SEQUEST using Proteome Discoverer 2.4 (Thermo Fisher Scientific Inc.) against a *Pseudotsuga menziesii* predicted proteins database from PineRefSeq project (54,830 entries, https://treegenesdb.org/FTP/Genomes/Psme/v0.5/ (accessed on 10 September 2023)). Spectra from peptides higher than 5000 Daltons (Da) or lower than 350 Da were rejected for quality reasons. The search parameters were as follows: mass accuracy of the monoisotopic peptide precursor and peptide fragments was set to 10 ppm and 0.6 Da, respectively. Only b- and y-ions were considered for mass calculation. Oxidation of methionines (+16 Da) and protein N-terminal acetylation (+42Da) were considered variable modifications, while carbamidomethylation of cysteines (+57 Da) was considered a fixed modification. Two missed trypsin cleavages were allowed. Peptide validation was performed using the Percolator algorithm [42], and only “high confidence” peptides were retained, corresponding to a 1% False Positive Rate at the peptide level. Peaks were detected, and Area Under the Curve was integrated using the Minora algorithm embedded in Proteome Discoverer. Proteins were quantified based on unique peptide intensities. Normalization was performed based on the total Douglas fir protein amount. The raw data are available in Appendix A, containing all the biological replicates and MS information.

A selection phase of the 11,563 proteins thus identified and quantified (Appendix A) was carried out according to the following two criteria: being of the Master protein type and being present in the three biological repeats of an organ. These 3462 filtered proteins are presented in Appendix A.

### 2.5. Gene Ontology (GO) Analysis

GO analysis was conducted using the Sma3s software [43] version 2. Sma3s is a tool for massive annotation of new protein sequences. It is composed of three integrative modules that annotate unknown sequences. These modules use a preliminary exhaustive BLAST search as their starting point and together generate annotations. We used the reference database UniRef90 based on the UniProt database (https://www.uniprot.org (accessed on 10 September 2023)) [44] and the “goslim” parameter that gives slim GO terms, which represent more generic annotations. Relocation of some classes of GO terms was carried out with custom scripts to simplify interpretation (see Appendix A for details). The presence of each protein in each organ was counted under the “Molecular Function” and “Biological Process” GO terms, allowing us to obtain the frequency of each GO term for each organ.

### 2.6. Statistical Analysis

Multivariate analyses were performed in R (https://www.cran.R-project.org (accessed on 10 September 2023)) using the mixOmics package [45] version 6.20.0.

## 3. Results

### 3.1. A wide Diversity and Abundance of Proteins Observed in All Organs

Proteomic analysis identified 3975 different proteins from Douglas fir organs, of which 3462 were quantified (see Appendix A and raw data in Appendix A), revealing a large disparity in protein presence and abundance between organs. Some samples are tissues and not organs, but for ease of understanding, organ names will be assigned generically below. Additionally, conifers differ from woody flowering plants in several ways, including reproductive structures that are cones rather than flowers. At the stage of collection, they are immature and then correspond to strobili.

For example, only 1178 proteins were identified in mature seeds, compared to 2358 in the xylem, 3246 in the mature somatic embryo, and 3264 in the immature somatic embryo (Figure 1a). Organ-specific differences in protein abundance are also visible in Figure 1b, in which the dendrogram, which classifies organs according to the abundance of proteins present, highlights two groups of proteomes: those with less than 3000 proteins (mature seed, xylem, immature seed, and needle) and those with approximately 3200 proteins. About 1100 proteins are present in all organs, although at varying levels of abundance. These proteins are most likely housekeeping proteins necessary for the functioning of cells. In contrast, 79 proteins were present only in a single organ (Figure 1c), mainly in the mature somatic embryo, but also in needles and male strobili. The analysis of 2-organ specific proteins allowed us to visualize proteomic similarities between organs. This is present between the immature somatic embryos and callus, as well as between the latter and the mature somatic embryos. Some similarity also appears between needles and stems, with 11 shared proteins that were not found in any other organ. Although functionally closely related, leaf buds and needles shared only three organ-specific proteins.

In a principal component analysis (PCA) performed with the mean of the biological repetitions of the 12 proteomes, a high percentage of variance (40%) was explained with just two components (PC1 = 23% and PC2 = 14%, Appendix A) Similar results can be observed between the 36 biological replicates, demonstrating good quality and repeatability of the analyses (Appendix A). The first component (PC1) separated samples according to the abundance of proteins. By this multivariate approach, several organs clustered into three major groups (Appendix A): (1) female strobilus/male strobilus/bud and root; (2) stem/xylem/needle/mature seed and immature seed; (3) mature somatic embryo/immature somatic embryo and callus. The second group of organs (mature seed, xylem, immature seed, xylem, and needle) seem to have more distinctive proteomes.

Several proteins were organ-specific (Appendix A), such as PSME_00018823-RA and PSME_00018769-RA (ribulose-1,5-bisphosphate carboxylase) in needles, PSME_00012089-RA (phospholipase D alpha 1-like) in buds, PSME_00028642-RA (aspartate aminotransferase) in immature seeds, PSME_00030113-RA (peroxidase 12) in the callus, and PSME_00031474-RA (chitinase-like) in xylem. PSME_00010512-RA (vicilin-like) seems specific to the mature somatic embryo, mature and immature seed. PSME_00037308-RA (endonuclease 4-like) was abundant in all organs except for needles, mature somatic embryos, immature seeds, and xylem and was very abundant in mature seeds (2 times more abundant than other organs). The proteomes of four organs—xylem, needles, and mature and immature seeds—were particularly protein-rich, containing more than twice the number of proteins of other organs.

### 3.2. Functional Specificities were Highlighted by GO Analyses

Gene Ontology (GO) functional annotation was applied to the 3462 quantified proteins (Appendix A), resulting in 31 Molecular Function (MF) GO terms, 51 Biological Process (BP) terms, and 33 Cellular Component (CC) terms (Appendix A). In addition, 59 Uniprot pathways were identified in this study.

In order to study the specificity of each proteome without a priori, we performed a PCA of the abundance of BP terms (Figure 2). In this analysis, the immature somatic embryo has been separated from other organs on PC1 due to proteins involved in the “DNA process“, “cell differentiation“, “cell mortality“, and “ribosome biogenesis“ (Figure 2b). On the other side of PC1, needles were correlated with proteins involved in “response to stress”, “carbohydrate metabolic process”, “energy”, and “secondary metabolic process”. The stem and xylem separated from other organs (Figure 2a) due to their positive correlation with proteins involved in the “biosynthesis process”, “small molecule metabolic process”, “cofactor metabolic process”, “amino acid metabolic process”, and “lipid metabolic process” (Figure 2b). Interestingly, some organs, like the mature seed’s position on the PCA performed on BP terms, matched its position on the PCA performed on MF terms (Appendix A) with a positive correlation with proteins involved in translation and negative correlations with proteins involved in “plasma membrane“, “cellular component organization“ and “cell wall organization, or biogenesis“.

### 3.3. First Analysis of Active Biological Processes in Douglas Fir Organs

The following BP terms were the most represented across all organs: “catabolic process”, “anatomical structure development”, “biosynthetic process”, “response to stress”, and “small molecule metabolic process” (Appendix A). These processes are indicative of strong cell growth activity supported by catabolic activity, as evidenced by the presence of “transport” and “cell cycle” BPs, as well as BPs related to protein synthesis, protein maturation, and energy supply (Appendix A). In contrast, BPs that were relatively infrequent across all organs were: “symbiotic relationships”, “cell motility”, “pigmentation”, “circulatory system process”, and “plasma membrane organization”. They attested to a more marginal activity, such as pigmentation.

These results could also be investigated by comparing the organs in which each BP was most abundant (Appendix A). When the abundance of any given BP was compared across organs, functional specificities of the organs emerged. For example, mature seeds generally had a much lower number of proteins involved in BPs than other organs. Conversely, the immature somatic embryo proteome had the most proteins involved in nearly every one of the BPs. Needles were among the organs with the most proteins involved in the energy process, which is likely directly related to their photosynthetic activity. The proteomes of leaf buds mature somatic embryos, and callus was abundant in GO terms related to organ development, such as “plasma membrane organization”, “lipid metabolic process” (intervening in the synthesis of plasma membranes), “cellular component organism”, and “anatomical structure development”.

The strong similarity in the proteomes of the callus, immature embryo, and mature somatic embryo (Figure 1b)—all of which are involved in vegetative multiplication—is linked to the functional similarity of these organs. The mature somatic embryos were nevertheless distinguished from the two other organs in accordance with their organ diversity, which is lacking in callus and immature somatic embryos. (Appendix A). The comparison between male and female strobili, which have very similar structures aside from their sexual organs, revealed a difference only in the abundance of the “DNA process” term.

## 4. Discussion

We compared the proteomes of 12 organs of Douglas fir, a tree of economic importance for timber production in North America and Europe. To our knowledge, this is the first study to analyze proteomic data with gene ontology annotation for multiple organs and characterization of the proteome profile of any tree species.

Our analysis identified 3975 proteins across all Douglas fir organs, comparable to the 4,000 proteins identified in the rice proteome [46] and the 2830 proteins identified in *Quercus ilex* [36]. Nevertheless, it is doubtful that this number captures all proteins, given that 36,000 coding genes are present in plants [47], whereas only 931 proteins are referenced in the Uniprot database for Douglas fir. This number of coding genes encompassing counterparts must be overestimated. Nevertheless, this discrepancy clearly illustrates the difficulty of extracting all proteins from an organ, as explained in a recent publication on this subject [48]. This technical difficulty is not limited to our study; however, we observe organ-specific protein counts quite similar to those obtained for other species: 3082 in the wood of *Populus tremula* [49], 1533 in the seeds of *Pinus radiata* [33] but only 629 in *Pinus patula* seeds [32]; 503, 1250 or 3000 for different stages somatic embryogenesis of *Larix principis-rupprechtii* [33], of *Picea glauca* [50] or of *Quercus suber* [51], respectively.

Global comparisons of the quantitative proteomics data allowed us to characterize the molecular specificity of some organs. Interestingly, the four organs (xylem, needles, mature and immature seeds (Figure 1a)) with less diversity of proteins show a list of very high and specific proteins, which may be explained by the function of the organs. For example, mature and immature seeds had lower overall protein diversity compared to other organs but with a very high abundance of specific proteins (e.g., vicilin-like protein), likely due to their role in storage [51,52]. The proportion of storage proteins was less important in the mature Douglas fir seeds compared to *Pinus occidentalis* seeds, where they were dominant [33]. However, a total of only 187 proteins were identified in the *P. occidentalis* seed proteome, compared to 1178 proteins in Douglas fir seeds, and storage proteins were still relatively abundant in Douglas fir seeds. The most abundant protein in Douglas fir seeds was endonuclease IV, an enzyme required for cell viability that can recognize and process some damaged nucleosides [53]. The abundance of this protein is likely attributable to its importance in post-germination nucleic syntheses. Similarly, Alfonso et al. (2020) [32] demonstrated the importance of proteins involved in oxidation-reduction reactions to seed preparation for germination. Finally, we observed much higher abundances of BP terms in the immature somatic embryo proteome than in the other organs (Appendix A), likely due to the totipotent nature of this type of tissue and its involvement in organogenesis. The overview of the main processes characterizing the 12 proteomes is summarized in Figure 3.

## 5. Conclusions

Here, we present an organ-specific map of the Douglas fir proteome, thereby revealing the localization of specific proteins along with their quantitative profiles across 12 organs. This study provides an important foundation for future analyses of organ-specific protein classes and supports transcriptomic and metabolomic studies seeking to better understand the specific functional biology of different organs of conifers and other trees.

## Figures and Tables

**Figure 1 biomolecules-13-01400-f001:**
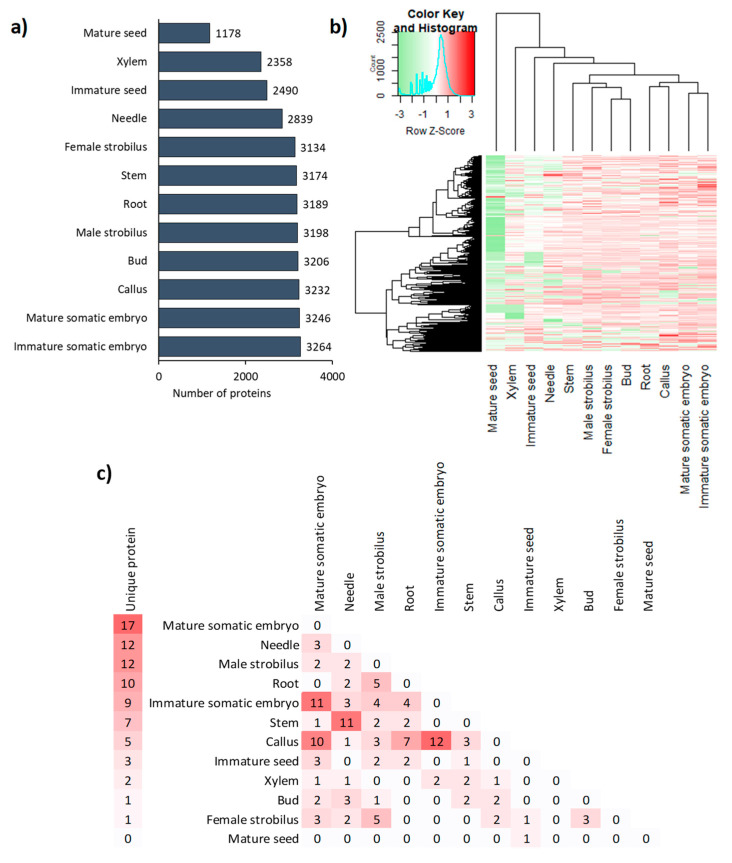
Specificity of protein abundances in the proteomes of 12 Douglas fir organs. (**a**) Bar plot of the number of proteins present in each organ. (**b**) Heatmap displaying log10-transformed counts of the quantifiable proteins for all organs. (**c**) Number of proteins present in only one organ (left side) or two organs (right side). For each organ, the presence or the average abundance of proteins is presented from 3 independent batches.

**Figure 2 biomolecules-13-01400-f002:**
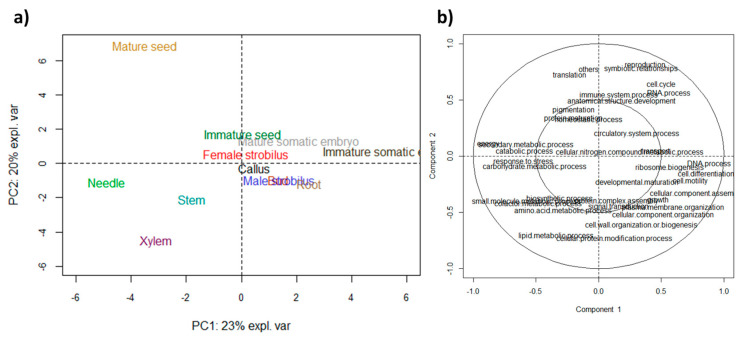
Comparison of the abundance of biological process GO terms of the proteomes of the 12 organs of Douglas-fir. Scaled Principal Component Analysis with (**a**) the individual plot and (**b**) the correlation circle plot associated with GO terms. Values for the x and y axes are those of PC1 and PC2, respectively.

**Figure 3 biomolecules-13-01400-f003:**
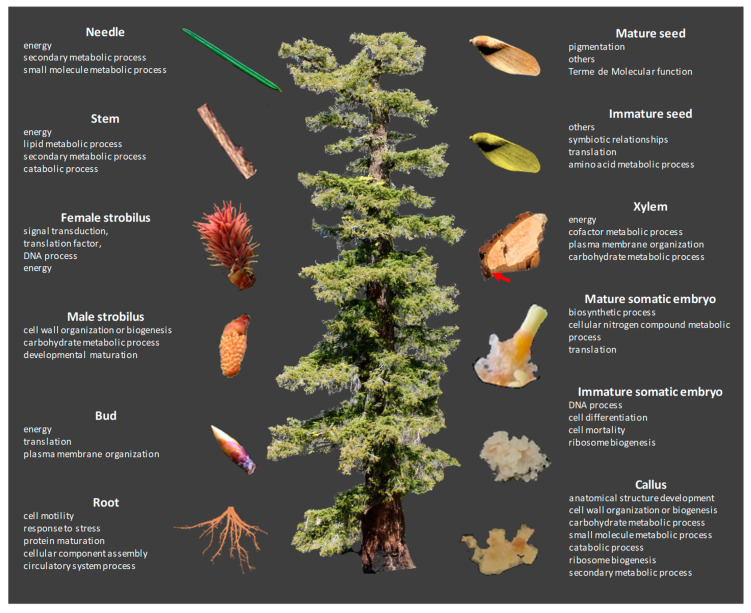
Global view of the major processes (Gene Ontology Biological Processes terms) for each analyzed organ.

## Data Availability

Not applicable.

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
