# Peer review of "Comprehensive Organ-Specific Profiling of Douglas Fir (Pseudotsuga menziesii) Proteome"

_biomolecules, 2023, doi:10.3390/biom13091400_

Round 1

Reviewer 1 Report

This is a short description on the proteome of the tree species Douglas fir. Although it is not an experimental study and, thus, does not bring any mechanistic insights, it brings relevant proteome and gene expression information on an orphan plant tree species. MS data generated can be further used for large-scale surrogates using different proteomic strategies.

I could not find any “red flags” regarding methodology. All data processing steps seem to be correctly carried out and properly described. Although of interest, there are still some points that need clarification/change before I can endorse this publication:

1. I could not find how this study helps genomic research. Authors used an EST database but could have explored genomic data from other public repositories, such as the TreeGenesDB. This would allow them to map proteins back to genomic sequencing and carry out a proteogenomic analysis in order to find novel coding sequences, for instance. Then, I could see the proteome data being used to unveil the genome complexity found in this species.

2. Quality of figures needs improvements. Figure 2A is redundant (similar to Figure 1B). Figure B is confusing and hard to visualize. Figure 3 lacks quality and brings no novelties.

3. As no GO enrichment test was carried out, differences between organs may be underestimated.

4. I guess readers would appreciate a website containing all proteome data (including abundances) organized and displayed in a friendly user interface. This is just a suggestion.

Author Response

Reviewer 1
This is a short description on the proteome of the tree species Douglas fir. Although it is not an experimental study and, thus, does not bring any mechanistic insights, it brings relevant proteome and gene expression information on an orphan plant tree species. MS data generated can be further used for large-scale surrogates using different proteomic strategies.
I could not find any “red flags” regarding methodology. All data processing steps seem to be correctly carried out and properly described. Although of interest, there are still some points that need clarification/change before I can endorse this publication:

1. I could not find how this study helps genomic research. Authors used an EST database but could have explored genomic data from other public repositories, such as the TreeGenesDB. This would allow them to map proteins back to genomic sequencing and carry out a proteogenomic analysis in order to find novel coding sequences, for instance. Then, I could see the proteome data being used to unveil the genome complexity found in this species.
      We thank you for these pertinent comments. We indeed used the PineRefSeq database hosted by TreeGenesDB. This detail has been added in the materials and methods section of the manuscript (line 137).  
      Additionally, we hope that you find the addition of motivations for the usefulness of this study in genomic research to be acceptable (lines 165-167).

2. Quality of figures needs improvements. Figure 2A is redundant (similar to Figure 1B). Figure B is confusing and hard to visualize. Figure 3 lacks quality and brings no novelties.
     Figures are in vectorial in the word version. We don’t know why the quality is degraded in the PDF file done by the MDPI system. This issue will be clarified with the editor.
We are sorry for the confusion, but figure 2A is a PCA carried out with GO abundance data, unlike      Figure 1B which is a heatmap which uses the abundance of all proteins. Legends of these figures have been clarified (lines 198, 231-233).
      We are aware that Figure 3 does not bring anything new here. But we remain convinced that this figure allows us to elegantly summarize our results in a way that gives better visibility of the specificity of each proteome. At a glance, the main GOs of each tissue are visible and easily comparable.

3. As no GO enrichment test was carried out, differences between organs may be underestimated.
     We chose to conduct multivariate statistical approaches in order to study and compare the specificity of GO terms for each organ. This approach has the advantage of not underestimating or overestimating the differences between organs and avoids to make choices of significance levels for each of the tests. Finally, we observe, via a PCA for example and the calculation of the principal component matrix, a synthesis of the information contained in each of these data sets without a priori.

4. I guess readers would appreciate a website containing all proteome data (including abundances) organized and displayed in a friendly user interface. This is just a suggestion.
      This is a great idea, and we thank you for proposing it. The creation of a web interface being an important work, we will consider creating a website with a user-friendly interface for our next studies in the future. Actually, the data have been deposited to the ProteomeXchange Consortium via the PRIDE partner repository, which make available to the readers after the publication of this manuscript.

Reviewer 2 Report

The authors used nLC- MS /MS to analyze proteomic data from 12 different organs of Douglas fir, a tree of economic importance for timber production. In addition, a number of important processes characterizing the 12 proteomes of selected organs were identified.

The work is written in the correct language. The methodology is exhaustively described. The results described are supported by detailed analyzes and visualizations. The conclusions drawn are accurate and the discussion is logical. The supplementary files are well prepared and facilitate the understanding of the results presented.

I did not find anything in the paper that aroused my reservations and I congratulate the authors for such a good research.

I can only ask the authors to correct/explain some doubts:

1. Line 96: CHAPS (capital letters)

2. This is confusing. In line 102 the authors write “Spots were incubated...", but in line 98-99 the authors write that they used SDS-PAGE electrophoresis and cut the protein bands from the gel to further isolate the proteins. Then how can "spots" (2D electrophoresis) appear in line 102?

3. From which database exactly were the reference sequences obtained? Please provide a link to this database. As the authors themselves mentioned, there are currently 931 entries (proteins) in UniProt for this species, and there are 58,905 nucleotide sequences in NCBI (18,142 EST). Could you please elaborate/explain this aspect?

4. This comment is related to the previous one. Accession numbers are references to which database? Should not the abbreviation of PSME be PSEM? (PSEudotsuga Menziesii)

5. Line 275: North American

6. Line 304: Authors should check references carefully. In this line, the authors describe studies by Alfonso et al, ref 32, but cite two items, 32 and 34. Why actually?

7. Funding: there is no direct grant number. Please add to this information.

8. I am also not sure about the nature of the article: short note, but this issue should be clarified with the editor.

After explaining the above notes, the article can be published in Biomolecules.

Author Response

Reviewer 2
The authors used nLC- MS /MS to analyze proteomic data from 12 different organs of Douglas fir, a tree of economic importance for timber production. In addition, a number of important processes characterizing the 12 proteomes of selected organs were identified.
The work is written in the correct language. The methodology is exhaustively described. The results described are supported by detailed analyzes and visualizations. The conclusions drawn are accurate and the discussion is logical. The supplementary files are well prepared and facilitate the understanding of the results presented.
I did not find anything in the paper that aroused my reservations and I congratulate the authors for such a good research.
I can only ask the authors to correct/explain some doubts:

1. Line 96: CHAPS (capital letters)
     This is done.

2. This is confusing. In line 102 the authors write “Spots were incubated...", but in line 98-99 the authors write that they used SDS-PAGE electrophoresis and cut the protein bands from the gel to further isolate the proteins. Then how can "spots" (2D electrophoresis) appear in line 102?
      This reviewer is right, this was a mistake. The text has been corrected on that way (line 105).

3. From which database exactly were the reference sequences obtained? Please provide a link to this database. As the authors themselves mentioned, there are currently 931 entries (proteins) in UniProt for this species, and there are 58,905 nucleotide sequences in NCBI (18,142 EST). Could you please elaborate/explain this aspect?
     The identification of proteins from mass spectrometry analysis have been done with the protein sequence file (54830 proteins; v.0.5) from the database of douglas fir available on TreeGenesDB. The link is now provided in the text in the 2.4 section (line 137).
     The large difference of number of sequences between Uniprot and NCBI could have many origins:
• Uniprot is a manually curated database, which allows the elimination of duplicates. Only existing identified proteins are recorded.
• NCBI records all EST sequences even if they correspond to non-coding sequences. Conifer genome has long repeated sequences with small sequence variations. A same codon can be codifying by many nucleotide sequences, leading to many different EST but only one unique protein.

4. This comment is related to the previous one. Accession numbers are references to which database? Should not the abbreviation of PSME be PSEM? (PSEudotsuga Menziesii)
     Protein identification was done only using the database indicated in the M&M (TreeGenesDB).      Thinking that this question from the reviewer could also appear to a reader, we modified the legends of tables 2S and 3S to provide this information.
     PSME is the prefix given to each accession number by the authors of the TreeGenesDB database. We assume it is PSeudotsuga MEnziesii.

5. Line 275: North American
     This has been corrected.

6. Line 304: Authors should check references carefully. In this line, the authors describe studies by Alfonso et al, ref 32, but cite two items, 32 and 34. Why actually?
       This was a forget to erase the old reference number. We checked that the further reference   number were not impacted. The text has been corrected.

7. Funding: there is no direct grant number. Please add to this information.
        We don’t have direct grant number, but only a link to the grand older (http://www.fondation.unilim.fr/sylvalim). We don’t think that this will be relevant to give it in the text.

8. I am also not sure about the nature of the article: short note, but this issue should be clarified with the editor.
         We apologize for this mistake, the article type is a “Brief report”

After explaining the above notes, the article can be published in Biomolecules.